# Mature tau pathology is not improved by interfering with interleukin-1 receptor signaling in two mouse models of tauopathy

Dylan J. Finneran[1], Brianna M. Jackman[1], Taylor Desjarlais[1], Alayna Henry[2], Ahlam S. Soliman[3], Patricia C. Muskus[4], Rama Shankar[5], Bin Chen[5,6,7], Kevin R. Nash[8], Dave Morgan[1]*, Marcia N. Gordon[1]

**1** Department of Translational Neuroscience, College of Human Medicine, Michigan State University, Grand Rapids, Michigan, United States of America, **2** Arizona College of Osteopathic Medicine, Midwestern University, Glendale, Arizona, United States of America, **3** Department of Neurodegenerative Science, Van Andel Institute, Grand Rapids, Michigan, United States of America, **4** Department of Pharmacology and Toxicology, Medical College of Wisconsin, Milwaukee, Wisconsin, United States of America, **5** Department of Pediatrics and Human Development, Michigan State University, Grand Rapids, Michigan, United States of America, **6** Department of Pharmacology and Toxicology, Michigan State University, East Lansing, Michigan, United States of America, **7** Department of Computer Science and Engineering, Michigan State University, East Lansing, Michigan, United States of America, **8** Department of Molecular Pharmacology and Physiology, University of South Florida, Tampa, Florida, United States of America

* scientist.dave@gmail.com

## Abstract

Prior work suggests that the cytokine interleukin-1β (IL-1β) may be a key regulator of tau pathology in the presence of amyloidosis. Here, we tested the possible benefits of interleukin-1 receptor antagonist (IL-1RA) gene therapy in two mouse models of tauopathy. We performed intracranial injections in the rTg4510 model, achieving approximately 300-fold over-expression in the hippocampus, and systemic injections in the PS19 model, resulting in approximately 10-fold over-expression. In neither model did we find substantial treatment effects with IL-1RA over-expression. We found large increases in *Il1b* gene expression in these mouse models, but considerably smaller increases in IL-1β protein. These data suggest that interleukin-1 receptor antagonist may not be a viable therapeutic strategy for pure tauopathies but cannot rule out possible benefits in amyloid-enhanced tauopathy, which appear to have larger elevations of IL-1β.

## Introduction

Alzheimer's disease (AD) is a progressive neurodegenerative disease with the neuropathological hallmarks of extracellular senile plaques, composed primarily of amyloid- beta (Aβ) peptide, and intracellular neurofibrillary tangles, composed of the tau protein. Activated microglia, the resident macrophages of the brain, are found

**Data availability statement:** ELISA, qPCR, and IHC data are available as supplementary information. Raw reads from RNA-sequencing are available from the NCBI Sequence Read Archive database (accession number PRJNA1228838).

**Funding:** This work was supported by Judy Fund and Alzheimer's Association Zenith Award (ZEN-15-321311) to MNG and AG055072, AG077651, and P30AG072931 to DM. The funding sources did not have a role in the study design; collection, analysis and interpretation of data; writing of the manuscript; or decision to submit for publication.

**Competing interests:** We have read the journal's policy and the authors of this manuscript have the following competing interests: MNG is a member of the Alzheimer's Association International Research Grant Program (IGRP) Council, for which she receives complementary membership in the Alzheimer's Association International Society to Advance Alzheimer's Research and Treatment (ISTAART) and registration for the Alzheimer's Association International Conference (AAIC). DM is supported by research funds from the Alzheimer's Association Zenith Award, the NIH grants R01AG077, R01AG055072, R01 AG062217, R01 AG 051500, P30AG072931, and Danaher Inc. DM is on the scientific advisory boards of SynapsDx, MindImmune and InMed pharmaceuticals. DM collaborates with BrightMind Biosciences. BC receives research funding from NIH R01 GM134307, NIH R01 GM145700, NIH R01CA286189, DOD HT94252410133, Corewell-MSU Alliance, HFH-MSU Alliance, and MSU Foundation. These funders had no role in the results presented in this manuscript.

surrounding Aβ plaques in both human AD brain and mouse models [1]. Recent genome wide association studies have implicated microglia in the pathogenesis of AD, by identifying that gene variants associated with increased risk for AD are expressed predominantly by cells of myeloid lineage, including microglia [2,3]. Therefore, it is hypothesized that the inflammatory response of microglia to Aβ deposits induces the hyperphosphorylation of tau in neurons [4].

Critical mediators of microglial activation are pro-inflammatory cytokines, such as interleukin-1 beta (IL-1β). IL-1β is produced as a precursor protein and must be cleaved by caspase-1 into its mature form to be active [5]. Mature IL-1β binds to the interleukin-1 receptor (IL-1R) complex, which consists of the ligand binding chain (IL-1R1) and the accessory chain (IL-1R3). In addition to IL-1β, interleukin-1 alpha (IL-1α) and interleukin-1 receptor antagonist (IL-1RA) are also ligands for IL-1R1 and all three have similar affinity for the receptor. Binding of either IL-1β or IL-1α causes association of IL-1R1 with IL-1R3 and signaling through Myd88 leading to p38 and JNK signaling as well as activation of the transcription factor NF-κB. IL-1RA regulates IL-1R1 signaling by inhibiting the association of IL-1R1 with IL-1R3 [6]. While IL-1α is constitutively expressed, mature IL-1β can be secreted by microglia upon activation and has been linked to AD pathogenesis. IL-1β is elevated in cerebrospinal fluid and there is a significant increase in IL-1β expressing microglia in brain of AD patients [7,8]. Studies in model systems have examined the effects of IL-1β on the hallmark pathologies of AD.

Over-expression of IL-1β in mouse models of amyloidosis significantly reduced Aβ plaque pathology [9]. However, in the 3xTg model that develops both amyloid and tau pathology, IL-1β over-expression significantly increased tau phosphorylation while reducing amyloid burden [10]. Systemic treatment with an IL-1 receptor blocking antibody significantly reduced phospho-tau while having minimal effects on amyloid burden in this model [11]. *In vitro*, microglia can be stimulated to release IL-1β by Aβ or other pro-inflammatory insults, which in turn causes phosphorylation of tau in primary neurons [12–14]. Treatment with IL-1RA can ameliorate this effect. In models of primary tauopathy, studies have demonstrated that IL-1β can exacerbate tau pathology. Conditioned media from *Cx3cr1⁻/⁻* microglia induced phosphorylation of tau in primary neurons from hTau mice and this effect could be blocked by pre-treatment with recombinant IL-1RA [13]. Similarly, adoptive transfer of *Cx3cr1⁻/⁻* microglia from hTau mice into nontransgenic mice induced tau phosphorylation, which could be blocked by administering recombinant IL-1RA [14]. These studies suggest that IL-1β may play a role in AD pathogenesis.

Recently, it was reported that disruption of the NLRP3 inflammasome, which reduces conversion of pro-IL-1β to mature IL-1β, among other actions, ameliorates the phenotype of both mouse models of amyloidosis [15] and models of tauopathy [16]. However, the precise downstream mechanism by which NLRP3 disruption alleviates tauopathy is not known. No study has examined if disrupting IL-1R signaling is beneficial in a transgenic model of pure tauopathy. Given that a recombinant human IL-1RA is approved for use in human peripheral inflammatory conditions (anakinra), we thought it feasible that increasing IL-1RA, and thus reducing IL-1 signaling

through IL-1R, might be a rational approach to moderating inflammation in AD. To address this, we designed and carried out studies in which we over-expressed IL-1RA using recombinant adeno-associated virus (rAAV) in two different mouse models of tauopathy, the rTg4510 mouse and PS19 mouse [17,18]. Activation of the NLRP3 inflammasome has been reported in both of these models [19–21]. Over-expression of IL-1RA was chosen as the preferred delivery method due to the long duration of the study and the limited fraction of peripherally administered IL-1RA that crosses the blood-brain barrier [22]. Furthermore, central administration of rAAV over-expressing IL-1RA limits the treatment effect to central immune cells, removing a potential confound of peripheral immune activation influencing pathogenesis in the chosen mouse models. In the rTg4510 model, we achieved a large degree of over-expression which was restricted to the hippocampus and anterior cortex. In the PS19 model, we achieved a modest degree of over-expression throughout the brain by administering the rAAV intravenously. In each model, we examined the resulting tau phenotype, hypothesizing that inhibiting IL-1 signaling would reduce tau hyperphosphorylation, aggregation and deposition, and the associated microglial response.

## Methods

### Animals

Animal use followed guidelines in the National Resources Council "Guide for the Care and Use of Laboratory Animals." All experiments were veterinary reviewed and approved by the Michigan State University Institutional Animal Care and Use Committee under protocol number 01/18-004-00. Pain and distress to experimental animals was reduced through the use of pre- and post-surgical analgesia and by the use of anesthetics during surgeries. Parental mutant tau (P301L) and tetracycline-controlled transactivator protein strains were maintained separately and bred to produce rTg4510s and littermate nontransgenic (NonTg) control mice as previously described [17]. PS19 mice and C57BL/6J mice were purchased from Jackson Laboratory (Strains 008169 and 000664, respectively). Genotype of experimental mice was determined from ear punch biopsy and confirmed after euthanasia by Transnetyx, Inc (Cordova, TN). Mice were singly housed prior to viral vector injection. The light cycle was maintained at 12-hour light/dark and mice were given food and water *ad libitum*.

### Cloning & adeno-associated virus production

Mouse *Il1rn* (IL-1RA) cDNA, including the endogenous secretion signal, was ordered as a gene block from Integrated DNA Technologies with a 5' SpeI sequence and 3' XhoI sequence. Insert and vector, pTR MCS-IG with the hybrid cytomegalovirus enhancer/chicken beta-actin promoter and a green fluorescent protein (GFP) reporter downstream of a polyoma virus 1 internal ribosome entry site (IRES), were digested with SpeI and XhoI and ligated to generate pTR IL-1RA-IG. The presence of the insert and terminal repeats was verified by digestion with SmaI. The sequence was verified by sequencing reaction and expression was confirmed by polyethylenimine transfection in HEK 293T cells purchased from ATCC (catalog number CRL-3216). The GFP control virus, pTR GFPW, was generated as described previously [23]. These vectors contain AAV2 terminal repeats, the ubiquitous CAG promoter, and bovine growth hormone poly-A signal for transcription termination. The control vector contained the woodchuck hepatitis virus posttranscriptional regulatory element for enhanced expression while the *Il1rn* containing vector had an IRES-GFP reporter. Recombinant AAV serotype 9 (rAAV9) particles were generated using the triple transfection method as described previously [24]. Viral titers were assessed by qPCR as described in [25] using primers against the IRES (5' – GGCTAGCCCCGGGGATGCA; 3' – ACGACATCACCGGGGAAACAGA) and WPRE (5' – GGCTGTTGGGCACTGACAAT; 3' – CCGAAGGGACGTAGCAGAAG). These vectors were used for injection intracranially into rTg4520 mice.

For systemic injection into PS19 mice we produced rAAV-PHP.eB vectors [26]. To restrict expression to neurons, the CaMKIIa promoter was excised from pTR C-MCSW using EcoRI and SpeI and ligated into pTR IL-1RA-IG. The corresponding control vector, pTR C-MCS-IG, was generated by digesting CaMKIIa promoter with EcoRI and NheI and ligated into pTR MCS-IG. These vectors contain AAV2 terminal repeats, polyoma virus 1 internal ribosome entry site,

and the bovine growth hormone poly-A signal for transcription termination. Recombinant PHP.eB AAV particles were generated using the triple transfection method [24]. The construct encoding PHP.eB (pUCmini-iCAP-PHP. eB) was a generous gift from Viviana Gradinaru (Addgene plasmid # 103005; http://n2t.net/addgene:103005; RRID: Addgene_103005) [26]. Viruses were titered by the dot-blot method using a non-radioactive biotinylated probe against GFP generated by PCR.

## Cell culture

An IL-1β reporter cell line, which secretes alkaline phosphatase upon treatment of IL-1β, was purchased from InvivoGen (Cat. No. hkb-il1bv2) and grown per the manufacturer's instructions. Cells in a 6-well plate, transfected with 1.5 µg of plasmid DNA encoding IL1RA using polyethylenimine, and grown for 24 hours. The cells were then lifted off the plate and approximately 75,000 cells were seeded per well of a 96-well plate in four replicates and allowed to grow for three days. Cells were then treated with either 100 ng/mL of recombinant IL-1β or PBS (n = 2) and incubated overnight. A 50 µL aliquot of media was assayed for alkaline phosphatase activity per the manufacturer's instructions. Cell viability was assessed using an MTT assay following the manufacturer's instructions (Sigma Aldrich, Cat. No. 11465007001).

## AAV injection

Three-month-old rTg4510 mice were randomly assigned to receive 2 µL per site of 1x10^{13} vg/mL of either AAV9-GFPW (n = 12, 8 male and 4 female) or AAV9-IL-1RA-IG (n = 13, 7 male and 6 female) unilaterally into the right anterior cortex (coordinates from bregma: 2.2 mm lateral, 2.2 mm anteroposterior, −3.0 mm vertical) and hippocampus (coordinates from bregma: 2.7 mm lateral, −2.7 mm anteroposterior, −3.0 mm vertical) with a flow rate of 0.5 µL/min [27]. Mice received 3.25 mg/kg of extended-release buprenorphine 30 minutes prior to surgery. Surgery was performed under isoflurane anesthesia with oxygenation. Four months after injection, tissue was collected. Mice were weighed and injected with a pentobarbital (100 mg/kg) and phenytoin (12.5 mg/kg) solution. The deeply anesthetized mice were transcardially perfused with 25 mL of 0.9% saline. The anterior pole of the brain was removed, separated into ipsilateral and contralateral hemispheres, and samples were snap frozen on dry ice. Cerebellum was also removed and snap frozen on dry ice. The remaining caudal portion of the brain (containing hippocampus) was fixed in 4% paraformaldehyde for 24 hours at 4°C.

Six-month-old PS19 mice were randomly assigned to receive 100 µL of either PHP.eB-MCS-IG (n = 14, 7 males and 7 females) at 3x10^{13} vg/mL or PHP.eB-IL-1RA-IG (n = 8, 3 males and 5 females) at 8x10^{13} vg/mL into the retro-orbital sinus under isoflurane anesthesia. Three months after injection tissue was collected. Mice were weighed and injected with a pentobarbital (100 mg/kg) and phenytoin (12.5 mg/kg) solution. The deeply anesthetized mice were transcardially perfused with 25 mL of 0.9% saline. Right hemibrain was dissected into anterior cortex, posterior cortex, hippocampus, and cerebellum which were snap frozen on dry ice. Left hemibrain was fixed in 4% paraformaldehyde for 24 hours at 4°C.

Fixed brain tissue was cryoprotected in sucrose by incubations in 10%, 20%, and 30% sucrose for 24 hours each. Brains were sectioned on the freezing stage of a sliding microtome horizontally into 25 µm thick sections. Sections were stored in PBS with 10 mM sodium azide at 4°C.

## Tissue homogenization, ELISA, & western blotting

Frozen brain samples were weighed and homogenized in RIPA buffer (10 µL per mg wet tissue mass) with protease inhibitor cocktail (Sigma Aldrich, Cat. No. P8340), and phosphatase inhibitor cocktails II & III (Sigma Aldrich, Cat. Nos. P5726 & P0044 respectively). The homogenate was sonicated (3 x 3-second pulses) and centrifuged at 50,000 x g for 1 hour at 4°C. The supernatant was transferred to a new tube, assayed for total protein by Pierce BCA (ThermoFisher, Cat. No.

23227), and frozen. The pellet was sonicated (3 x 3-second pulses), resuspended in 70% formic acid (2 µL per mg wet tissue mass) for 1 hour at room temperature, neutralized in Tris, and assayed for total protein by BCA, and frozen.

Invitrogen ELISA kits were used to analyze human total tau (ThermoFisher, Cat. No. KHB0041), pSer396 phospho-tau (ThermoFisher, Cat. No. KHB7031), pSer199 phospho-tau (ThermoFisher, Cat. No. KHB7041). Oligomeric tau was assayed using the TOC1 ELISA as described in [28]. Commercially available kits were purchased for IL-1RA (ThermoFisher Cat. No. MRA00), IL-1β (R&D Systems Cat. No. MLB00C-1), PSD95 (LSBio Cat. No. LS-F7142), and synaptophysin (LSBio Cat. No. LS-F6345). Prior to ELISA analysis, supernatant and formic acid samples were diluted in 1X PBS and sample buffer to be within standard curve range of each respective ELISA. Assays were performed per manufacturer's instructions.

For Western blotting, 10 µg of supernatant was diluted into Laemmeli sample buffer (Bio-Rad Cat. No. 1610747) with BME and boiled for 10 minutes at 95°C. Samples were loaded into a 4–20% TGX Stain Free Gels (Bio-Rad Cat. No. 5678095) and separated by electrophoresis. Gels were activated per manufacturer's instructions, transferred onto nitrocellulose, and blots were imaged for total protein. The blots were then blocked with 5% milk in TBS for 1 hour at room temperature and incubated in rabbit anti-pTau (Anaspec Cat. No. AS-54963) at 1:1000 in 5% milk in 0.1% TBS-Tween overnight at 4°C. Blots were washed 3x10min in 0.1% TBST and probed with anti-rabbit secondary antibody (LiCor Bio Cat. No. 926–32211) at 1:20,000 in 0.1% TBST for 2 hours at room temperature. Blots were washed 3x10min in 0.1% TBST and imaged. Blots were analyzed using Bio-Rad Image Lab. Band of interest was normalized to total protein signal.

## Immunohistochemistry

Six to eight evenly spaced horizontal sections spanning the brain were chosen for analysis. Immunohistochemistry experiments were performed as described previously [29]. Commercially available primary antibodies were purchased and optimum dilution determined empirically including negative control tissue. Polyclonal chicken anti-GFP (Abcam Cat. No. ab13970) was used at 1:30,000. Biotinylated mouse monoclonal anti-human phospho-tau AT8 (ThermoFisher Cat. No. MN1020B) was used at 1:10,000 for chromogenic staining and 1:3,000 for fluorescent staining. For chromogenic immunostaining, floating sections for each animal were placed into a multi-sample staining tray. Endogenous peroxidases were blocked (10% methanol, 3% hydrogen peroxide in PBS) and tissue was permeabilized (0.2% lysine, 0.1% Triton X-100, 4% goat serum in PBS). Sections were incubated at room temperature overnight in the appropriate primary antibody in 4% goat serum in PBS. Sections were washed 3x in PBS and incubated with biotinylated secondary antibody (1:3,000) in 4% goat serum in PBS for 2 hours at room temperature. Following three PBS washes, the sections were incubated with Vectastain Elite ABC kit for enzyme conjugation for one hour at room temperature. Sections were stained with 0.05% diaminobenzidine with nickel enhancement and 0.03% hydrogen peroxide for five minutes at room temperature. For fluorescent immunostaining, sections were permeabilized and then incubated in primary antibody overnight as described above. Sections were then washed three times in PBS and incubated with streptavidin conjugated to Alexa 594 (ThermoFisher Cat. No. S32356; 1:3,000). Each assay omitted some sections from primary antibody incubation to evaluate nonspecific binding of the secondary antibody. Fluorescently labelled sections were washed three times in PBS, mounted onto slides, and allowed to dry overnight. Sections were then stained with DAPI (1:3000 in PBS; ThermoFisher Cat. No. 62248) for 5 min and washed briefly in PBS before being cover slipped with ProLong Glass (ThermoFisher Cat. No. P36980). Chromogenic sections were mounted onto slides, dehydrated, and cover slipped. Slides were digitized using the Zeiss AxioScan.Z1 scanning microscope. NearCyte IAE software (created by Andrew Lesniak) used hue, saturation, and intensity to segment the images and identify threshold settings using the lightest and darkest section. Once identified, these values were held constant for analysis of every section labeled with that stain. For fluorescent stains, only intensity was used to segment images. The percent area staining was determined mathematically by dividing the number of segmented pixels by the total number of pixels in the region of interest.

## Statistical analysis

Biochemical and histological data were analyzed by ANOVA, two-way or one-way as appropriate, with Tukey's post-hoc test or two-tailed Student's t-test. Linear correlations were performed using Pearson r. Results were assumed significant if p < 0.05. Analyses were performed using GraphPad Prism version 10.4.1.

## Nucleic acid isolation, qPCR, and RNASeq

Genomic DNA and RNA were isolated simultaneously from posterior cortex using Qiagen's "AllPrep" Kit per the manufacturer's instructions. RNA was quantified by nanodrop. RNA was reverse transcribed to cDNA using Bio-Rad iScript RT and the relative expression of mouse *Il1b* RNA was determined by quantitative PCR using the relative standard curve method. In rTg4510 mice, pre-made primer set for *Il1b* was ordered from Integrated DNA Technologies (assay no. Mm.PT.58.41616450) and custom primers for the housekeeping gene *18S* (Forward – GTA ACC CGT TGA ACC CCA TT; Reverse – CCA TCC AAT CGG TAG TAG CG). Analysis of a melting curve after amplification showed only one peak. In PS19 mice, primer/probe sets were ordered from ThermoFisher: *Il1b* assay number Mm00434228_m1, *Gapdh* assay number Mm99999915_g1. For RNA sequencing, twelve samples (n = 6) were digested with DNase I (NEB Cat. No. M0303S) and the RNA purified via spin column kit (NEB Cat. No. T2040) before being sent for total RNA sequencing. Samples were sequenced using 100 bp paired-end reads with at least 50 million reads per sample. Sequencing reads were mapped to the M53 transcriptome using the ENSEMBL GRCm39 annotation with the STAR aligner under default parameters [30]. One outlier was identified with a principle component analysis and excluded from the downstream analysis, resulting in n = 5 AAV-GFP treated mice. Differentially expressed (DE) genes were quantified using the edgeR package [31] with the criteria: log2 fold change ≥ 0.5 or ≤ −0.5 and adjusted P-value ≤0.20. DE genes were identified by comparing IL1Ra-treated mice to GFP-treated mice. The plots were visualized using ggplot2 in R (version 4.3.2).

## Results

### IL-1RA over-expression in rTg4510 mice

Increased expression of cytokines has been observed in AD patient brain and mouse models of AD-like pathology. Similarly, we observed an age-dependent increase in *Il1b* mRNA in the rTg4510 mouse model of tauopathy (Fig 1A) as well as an increase in IL-1β protein at six months of age (Fig 1B). To investigate if IL-1βsignaling contributes to tauopathy in mice, we constructed a vector over-expressing murine IL-1RA (Fig 2A). We confirmed this vector over-expresses a secreted and bioactive IL-1RA protein using a reporter cell line for IL-1βsignaling (Fig 1C). We observed a significant reduction of IL-1β signaling in cells over-expressing IL-1RA (Fig 1D) and without producing cytotoxicity in this assay (S1 Fig). This vector and a GFP-expressing control vector (Fig 2A) were packaged into rAAV9 and injected unilaterally into the hippocampus and into the anterior cortex of three-month-old rTg4510 mice. After four months of expression, we observed an approximately 300-fold over-expression of IL-1RA as measured by ELISA in the ipsilateral anterior cortex of IL-1RA-injected mice compared to endogenous levels in AAV-GFP injected mice (analyzed by two-way ANOVA, significant interaction term [F(1, 28) = 7.539, p = 0.0104], Fig 2B) and approximately 70,000-fold excess IL-1RA compared to endogenous IL-1β (Fig 1B). The elevated expression could also be detected on the contralateral anterior cortex of IL-1RA injected mice, albeit at a roughly 10-fold elevation over endogenous levels. We further observed expression of the GFP reporter in the ipsilateral hippocampus of IL-1RA injected mice by immunohistochemistry, demonstrating successful transduction of this structure as well (Fig 2C).

### IL-1RA over-expression modestly reduces soluble tau oligomers

IL-1RA over-expression did not significantly modify RIPA-soluble total tau, pS199 tau, or pS396 tau as measured by ELISA (Fig 3A–3C). Oligomeric tau, measured using a TOC1 ELISA, was slightly reduced in IL-1RA-treated mice compared to GFP-treated control mice (analyzed by two-way ANOVA, main effect of treatment [F(1, 46) = 4.86, p = 0.03]; Fig 3D). The RIPA-insoluble tau fraction showed similar results as the soluble fraction, with no effect of IL-1RA

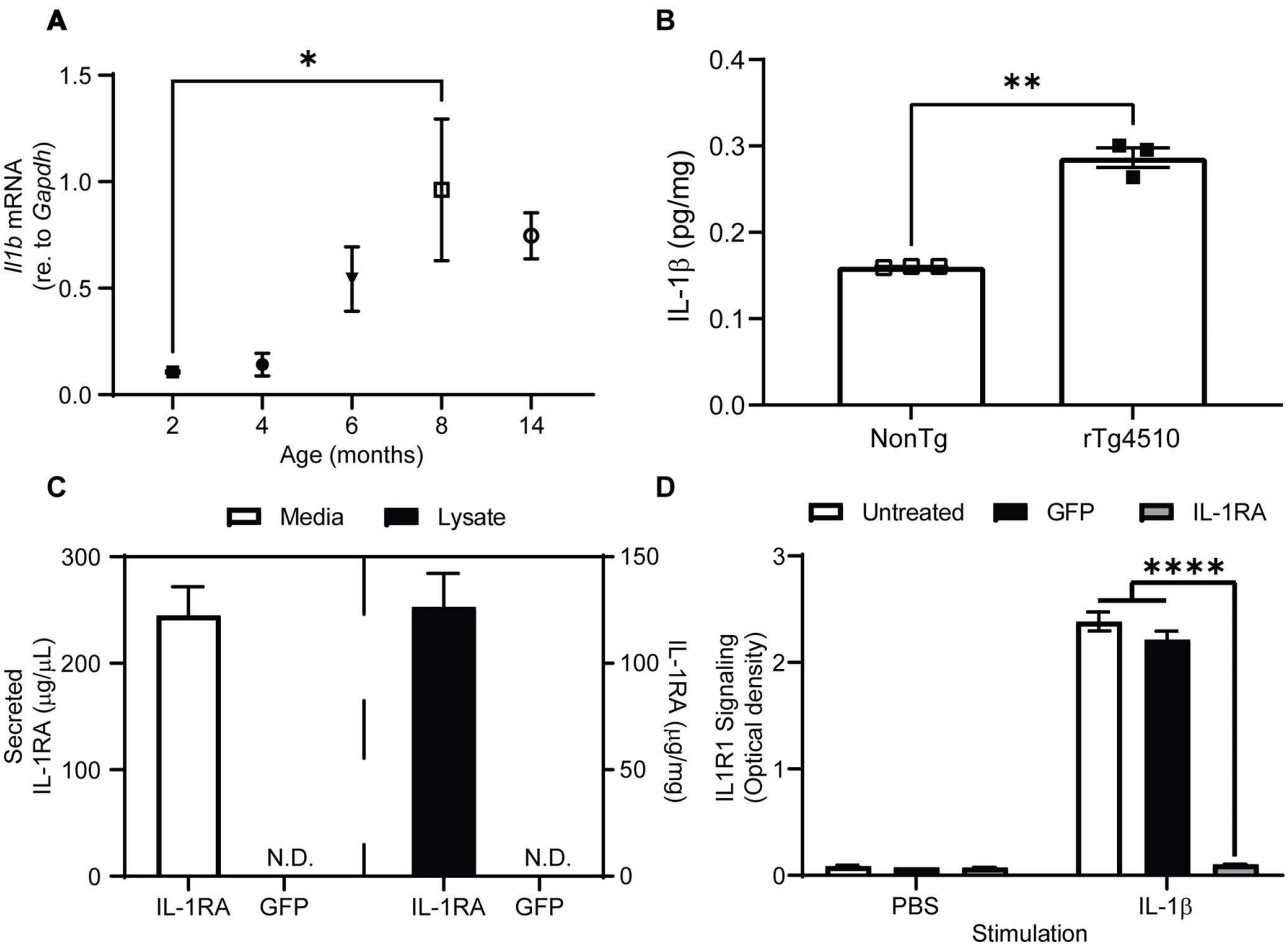

**Fig 1. Elevated *Il1b* message and protein in aged rTg4510 mice and *in vitro* expression and activity of IL-1RA expression vector. (A)** Graph of relative expression of *Il1b* in rTg4510 mice of different ages measured by qPCR. *Il1b* mRNA peaks at 8 months of age. * denotes p < 0.05 analyzed by one-way ANOVA & Tukey's post-hoc test; n = 4−6. **(B)** Graph of mouse IL-1β protein in RIPA-soluble supernatant of anterior cortex of four-month-old NonTg and six-month-old rTg4510 mice assayed by ELISA. ** denotes p < 0.01 by two-tailed Student's t-test; n = 3. **(C)** Graph of mouse IL-1RA secreted into the media (left panel) & in the cell lysate (right panel) of HEK 293T cells transfected with pTR IL-1RA or pTR GFPW (n = 3 wells) measured by ELISA. Mouse IL-1RA is detectable in both the cell lysate and secreted into the media 72 hours after transfection. **(D)** Graph of activity of secreted embryonic alkaline phosphatase in cell culture media. HEK-Blue IL-R cells transfected with pTR-IL-1RA do not respond to IL-1b stimulation. **** denotes p < 0.0001 analyzed by two-way ANOVA & Tukey's post-hoc test; n = 2. Similar results were observed in an independent experiment. Analyzed by two-way ANOVA; n = 2. Data presented as mean ± SEM. **N.**D. – not detectable.

over-expression total or pS396 insoluble tau levels (Fig 4A and 4C). For insoluble pS199 tau there was a main effect of hemisphere with significantly increased levels in the rAAV injected hemisphere (analyzed by two-way ANOVA, [$F_{(1, 46)}$ = 5.756, p = 0.0205]; Fig 4B). Examining phospho-tau burden histologically, we observed no effect of IL-1RA over-expression on AT8-positive tau (Fig 5A and 5B). These data indicate that while there may be a modest impact on soluble tau oligomers, IL-1RA over-expression does not modify tau burden in rTg4510 mice.

## Tau deposition occurs stochastically in rTg4510 cortex

Since there was no significant effect of IL-1RA over-expression on tauopathy, we pooled both treatment groups together to gain further insight into tau deposition in rTg4510 mice. We detected a high degree of concordance of soluble total tau

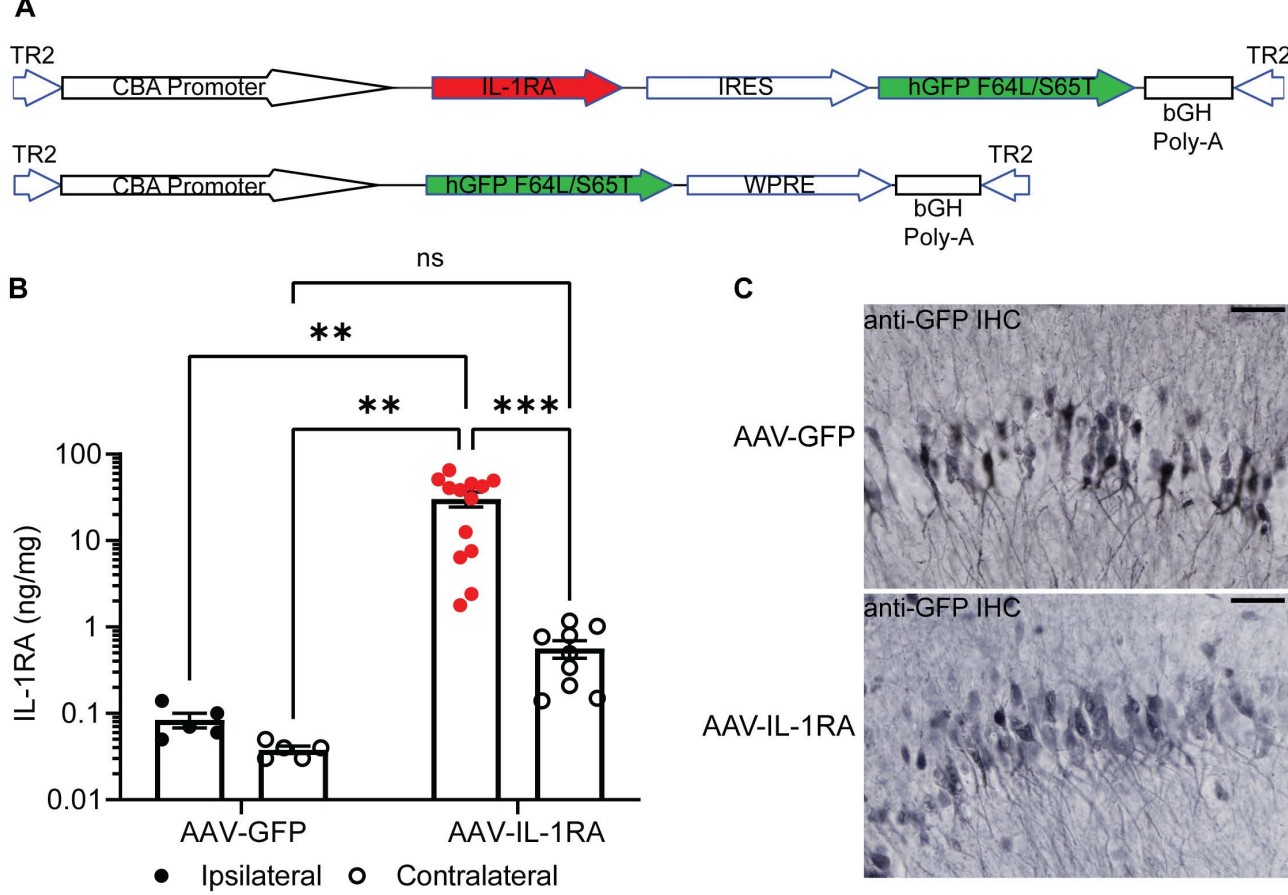

**Fig 2. Over-expression of mouse IL-1RA in mouse brain. (A)** Diagram of AAV genomes administered. AAV-IL-1RA contains polyoma 1 virus internal ribosome entry site and GFP reporter. AAV-GFP contains wood chuck hepatitis virus posttranscriptional regulatory element. **(B)** IL-1RA normalized to total protein concentration in the RIPA-soluble fraction of the frontal pole homogenate measured by ELISA. Animals injected with AAV-IL-1RA had significantly greater IL-1RA in the ipsilateral cortex than animals that received AAV-GFP. Note the Y-axis is presented as log scale. Analyzed by two-way ANOVA with Tukey's post-hoc test; *** denotes $p < 0.001$, ** denotes $p < 0.01$, "ns" denotes $p > 0.05$; n = 5−13. **(C)** Representative images of GFP expression in the ipsilateral hippocampi of animals that received AAV-GFP (top) and AAV-IL-1RA (bottom), showing similar transduction. Scale bar represents 200 mm. Data presented as mean ± SEM; points represent values of individual mice.

in the two hemispheres from the same mouse, as expected if gene dosage controlled the levels of expression (Fig 5C). However, there was no significant correlation between the two hemispheres for insoluble total tau, suggesting independent events differentially initiated tau aggregation at different times in the two hemispheres (Fig 5D). These data indicate that insoluble tau deposition likely begins in a stochastic manner independently in different brain regions.

### *Il1b* is elevated in PS19 mice

Concerned that the extensive, mature tauopathy in the rTg4510 mouse may be resistant to inhibition of IL-1R signaling, or that the restricted distribution of parenchymal injections might account for failure of IL-1RA to reduce tauopathy, we next tested IL-1RA in a less aggressive model of tauopathy, PS19 [18], using a systemically administered rAAV capsid we previously found had widespread and relatively uniform transduction throughout the brain (Fig 6C) [23]. We first documented a significant increase in *Il1b* mRNA (Fig 6A) and IL-1β protein (Fig 6B) in nine-month-old PS19 mice compared to NonTg

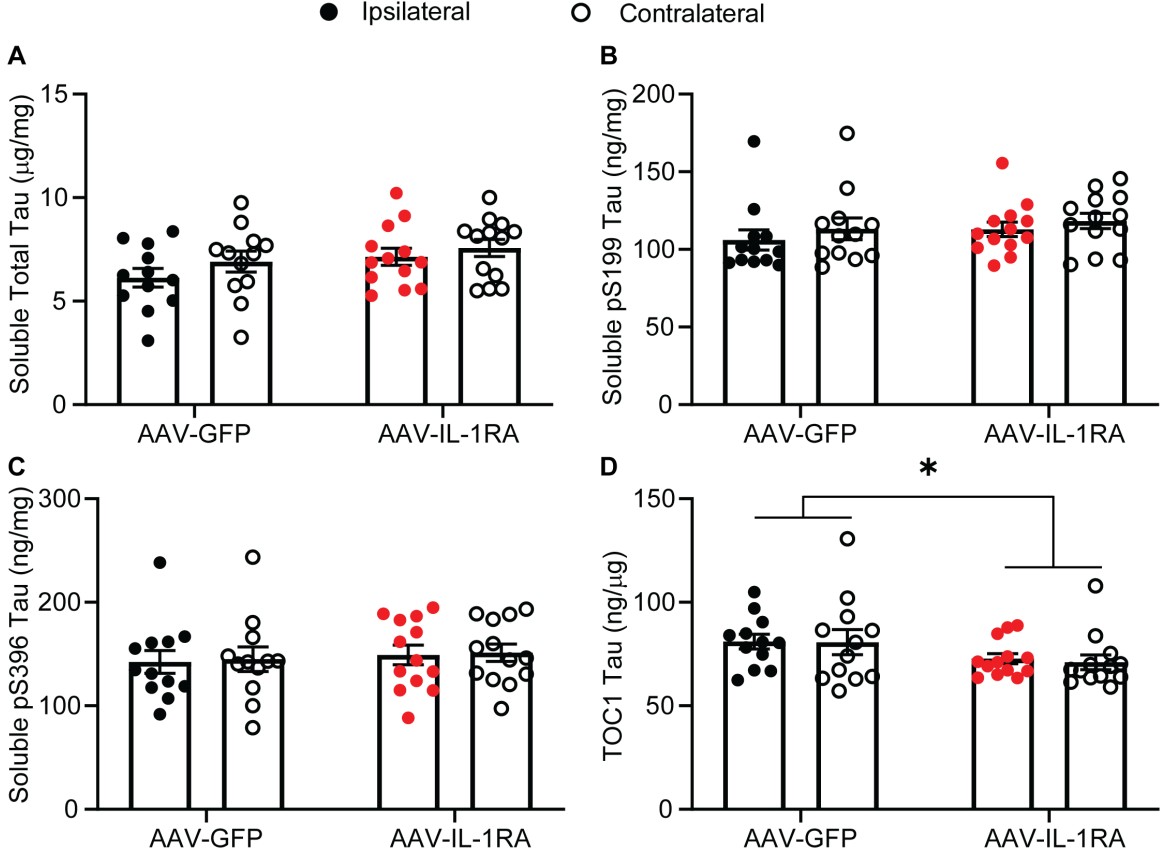

**Fig 3. ELISA for tau variants in the RIPA-soluble fraction. (A)** Total tau concentration. **(B)** Phospho-serine 199 tau concentration. **(C)** Phospho-serine 396 tau concentration. **(D)** TOC1-positive oligomeric tau concentration. Mice injected with AAV-IL-1RA had significantly less TOC1-positive tau than mice injected with AAV-GFP (main effect of treatment). Data normalized to total protein concentration and presented as mean ± SEM; points represent values of individual animals; n = 12-13. Analyzed by two-way ANOVA, * denotes p < 0.05.

mice in agreement with prior reports. We then constructed rAAV-PHP.eB vectors that cross the blood brain barrier, using a neuron specific promoter (CaMKIIα) and injected them intravenously into six-month-old PS19 mice. After three months of expression, we observed an approximately 10-fold over-expression of IL-1RA in mice injected with the AAV-IL-1RA vector compared to AAV-GFP-injected controls (Fig 6D).

## IL-1RA over-expression does not significantly alter tau burden in PS19 mice

IL-1RA over-expression did not significantly reduce RIPA-soluble total tau, pS199 tau, or pS396 tau measured by ELISA in the hippocampus of PS19 mice as compared to GFP-injected control mice (Figs 7A–7C and S2). Similarly, TOC1-positive soluble tau oligomers were unchanged in IL-1RA-injected mice compared to GFP-injected controls (analyzed by Student's t-test [t(21) = 0.75, p = 0.46]; Fig 7D). Examining RIPA-insoluble tau, we observed no significant changes in total tau, pS199 tau, or pS396 tau as measured by ELISA in IL-1RA-injected mice compared to GFP-injected controls (Fig 8A–8C). Finally, we observed no significant changes in AT8 histology in IL-1RA-injected mice compared to GFP-injected controls (Fig 9A–9C). These data indicate IL-1RA over-expression does not alter tau burden in PS19 mice.

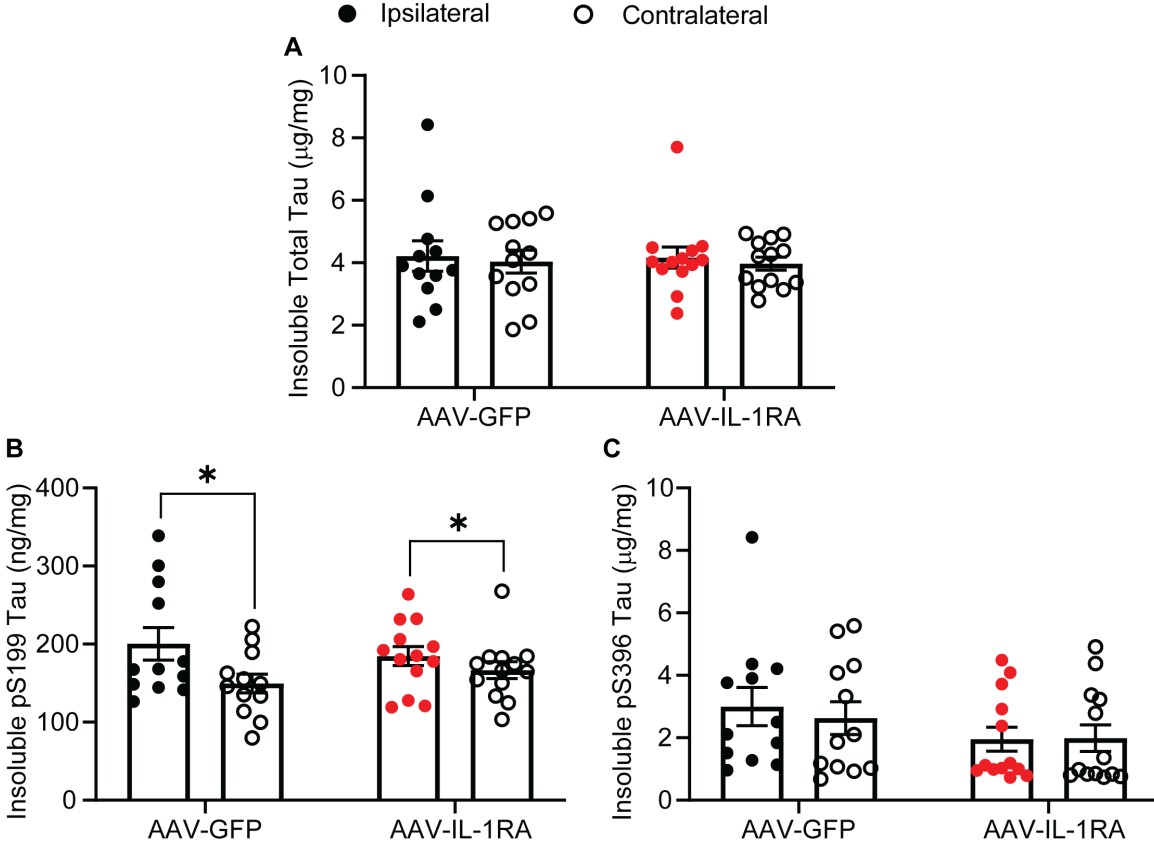

**Fig 4. ELISA for tau variants in RIPA-insoluble fraction. (A)** Total tau concentration. **(B)** Phospho-serine 199 tau concentration. We observed a main effect of hemisphere. **(C)** Phospho-serine 396 tau concentration. Data normalized to total protein concentration and presented as mean±SEM; points represent values of individual animals; n=12-13. * denotes p<0.05 by two-way ANOVA.

### IL-1RA over-expression alters innate immunity-associated gene expression in PS19 mice

To determine if there were any transcriptomic changes associated with over-expression of IL-1RA, we performed bulk RNA sequencing analysis of the posterior cortex. We confirmed a significant increase in normalized gene counts of *Il1rn* in IL-1RA-injected mice and observed significant reductions in MCH-II associated genes (*H2-Aa*, *H2-Ab1*, *Cd74*, Table 1). However, we did not observe a significant reduction in MCH-II immunoreactivity in IL-1RA over-expressing mice (Fig 10).

## Discussion

Here, we over-expressed IL-1RA in two models of primary tauopathy. In the rTg4510 mouse model, we achieved robust over-expression of IL-1RA observing a 300-fold increase over basal levels in the injected hemisphere. This corresponds to an approximately 70,000-fold excess of IL-1RA, by mass, compared to IL-1β, well above the estimated 10−100 fold excess required for 50% inhibition of IL1R1 signaling [32]. We observed no significant alterations in RIPA-soluble or -insoluble total tau or phospho-tau. We did observe a slight but significant reduction in RIPA-soluble TOC1-positive tau oligomers. We also examined tau pathology histologically by staining for AT8-positive tau. However, we did not observe a significant effect of increased IL-1RA. While some studies have published significant modifications of tauopathy using anti-inflammatory treatments in this model [33–36], our data suggest that IL-1RA may not be a successful therapeutic strategy through which to significantly impact disease progression in this model.

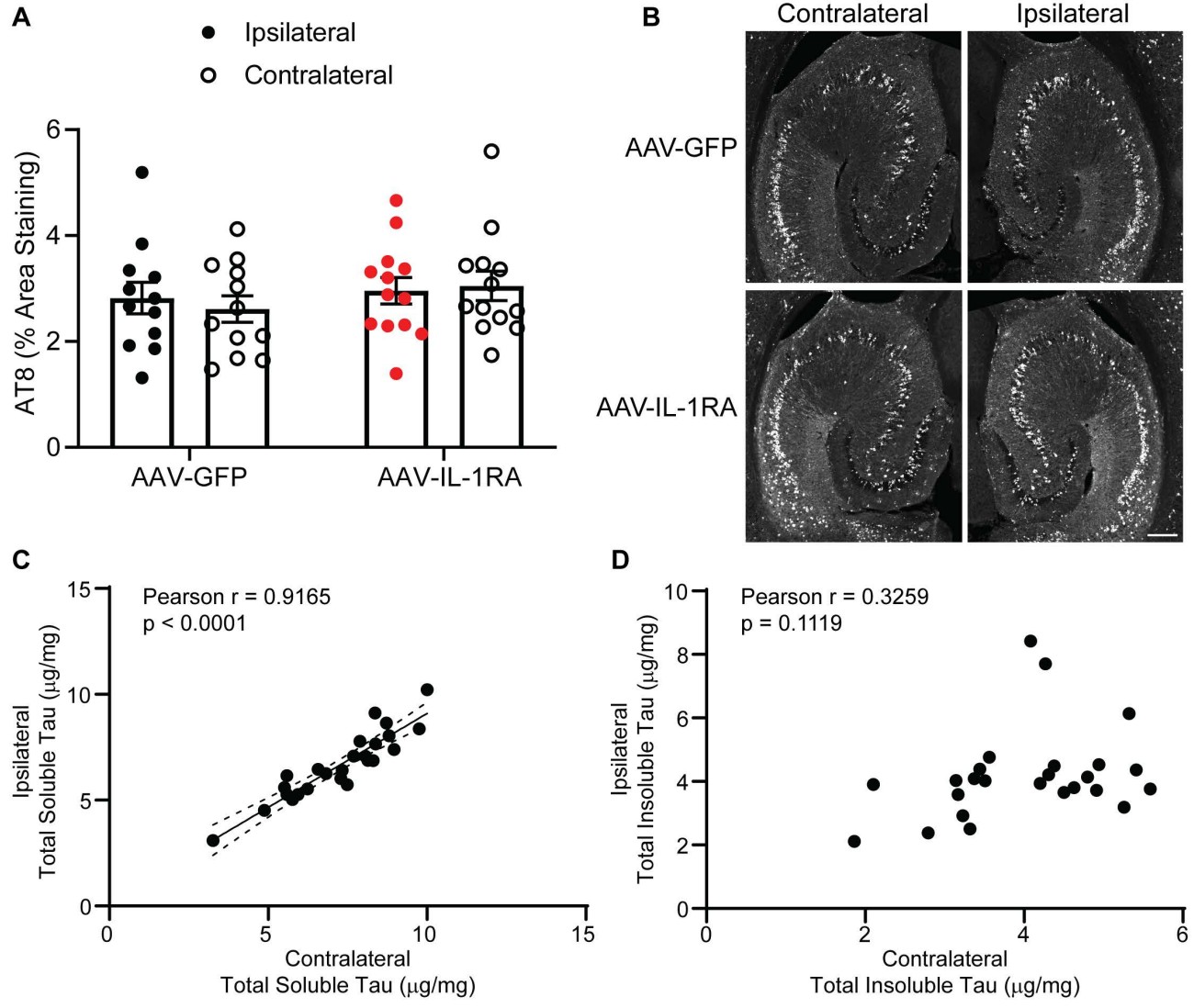

**Fig 5. AT8 is unaltered by IL-1RA overexpression and insoluble tau deposition may proceed in a stochastic manner in rTg4510 mouse brain.**
**(A)** Percent area staining of AT8 in the ipsilateral and contralateral hippocampi. Data analyzed by two-way ANOVA. **(B)** Representative images of AT8 immunofluorescence. Scale bar represents 200 mm. **(C)** Simple linear regression of RIPA-soluble total tau in the contralateral and ipsilateral hemispheres. There is a significant positive correlation between contralateral total tau and ipsilateral total tau. **(D)** Simple linear regression of RIPA-insoluble total tau in the contralateral and ipsilateral hemispheres. There is not a significant correlation between contralateral total tau and ipsilateral total tau. Data presented as mean ± SEM; points represent values of individual animals; n = 12-13.

We suspected that the failure of the data to support our hypothesis may be due to the aggressive nature of the rTg4510 mouse model used in our initial experiment. Thus, we turned to the less aggressive PS19 mouse model of tauopathy in which gliosis has been demonstrated to emerge prior to the onset of tau deposition, indicating that inflammation may play a more causal role in disease progression [18]. Additionally, we wanted to more uniformly transduce the area of the brain that over expressed IL-1RA and therefore chose to intravenously administer an AAV capsid capable of crossing the blood-brain barrier, thus transducing the entire CNS [26]. We achieved approximately 10-fold over-expression of IL-1RA compared to control-injected PS19 mice, resulting in an approximate IL-1RA to IL-1β mass ratio of 1,000, again above the

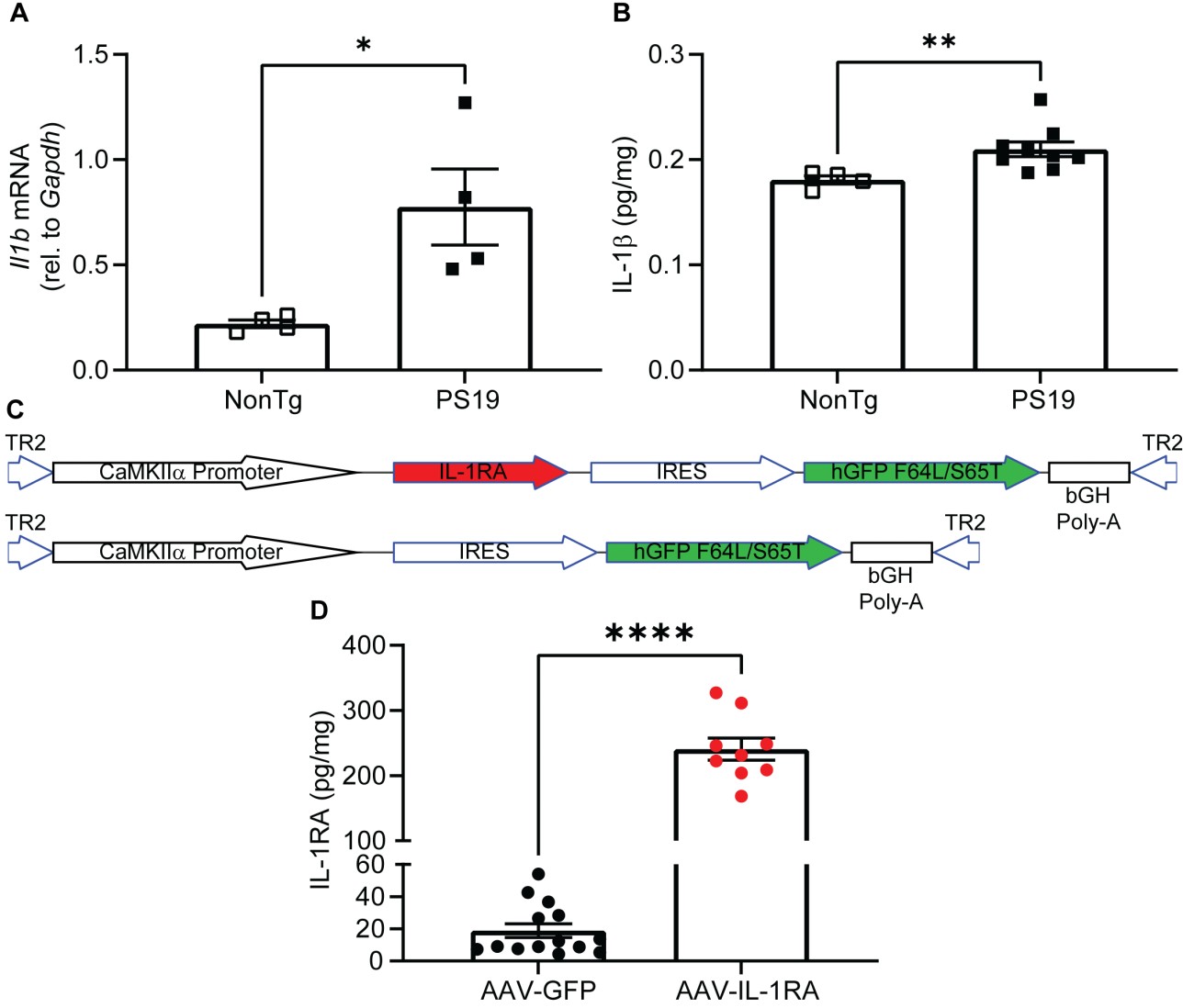

**Fig 6. *Il1b* is elevated in PS19 mice. (A)** Relative expression of *Il1b* mRNA relative to *Gapdh* in cortex of 9-month-old PS19 and NonTg mice measured by qPCR. *Il1b* message is significantly elevated in aged PS19 mice compared to NonTg mice (n = 4). **(B)** Graph of IL-1β protein measured by ELISA in anterior cortex of 9-month-old PS19 and NonTg mice. We observed a significant elevation of IL-1β protein in PS19 mice compared to NonTg mice (n = 4-9) **(C)** Diagram of AAV genomes administered systemically using PHP.EB capsid. **(D)** IL-1RA protein normalized to total protein concentration in hippocampus of AAV-GFP and AAV-IL-1RA injected mice measured by ELISA. Mice injected with AAV-IL-1RA had approximately 10-fold increase in IL-1RA protein (n = 9-14). Data presented as mean ± SEM; points represent values of individual mice. * denotes $p < 0.05$, **** denotes $p < 0.0001$ by two-tailed Student's t-test.

theoretical IC50. However, we again did not observe a significant effect on tau pathology. Total tau and phospho-tau levels in both RIPA-soluble and -insoluble fractions were unchanged by IL-1RA treatment.

Aside from its effect on tauopathy, we hypothesized IL-1RA would modulate the inflammatory phenotype. Therefore, we used an unbiased approach, bulk RNA sequencing, to examine any transcriptomic changes in PS19 mice resulting from IL-1RA over-expression. We observed significant reductions in MHC-II associated genes (Table 1), which is consistent with our hypothesis that IL-1RA would exert broad anti-inflammatory effects in brain. Interestingly, these genes are

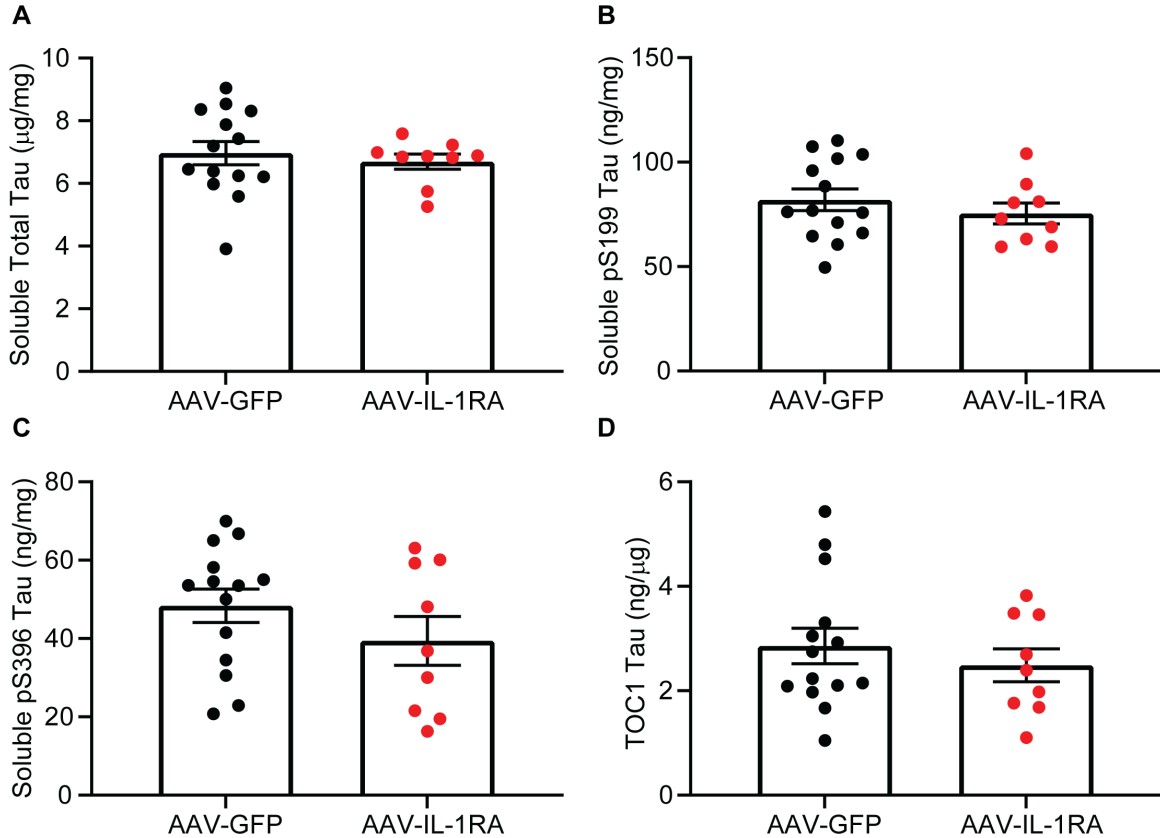

**Fig 7. IL-1RA over-expression does not modify soluble tau in PS19 mice.** Graph of RIPA-soluble total tau **(A)**, pS199 tau **(B)**, pS396 tau **(C)**, and TOC1 oligomeric tau **(D)**. IL-1RA over-expression did not modify RIPA-soluble tau measured by ELISA. Data normalized to total protein concentration and presented as mean ± SEM; points represent values of individual mice; n = 9-14. Analyzed by two-tailed Student's t-test.

reported as up-regulated in microglia from two different mouse models of amyloidosis, indicating that over-expression of IL-1RA may have altered the activation state of microglia in brain [37,38]. Thus, we conclude it is likely that IL-1RA was expressed and exerted bioactivity, albeit less robustly than we anticipated.

Taken as a whole, our data suggest that IL-1RA may not be a viable therapeutic strategy for treatment of tauopathies, possibly including AD. Reports implicating the NLRP3 inflammasome in the progression of tauopathy hint at a possible role of IL-1β, since cleavage of pro-IL-1β is downstream of NLRP3 activation [16,39,40]. However, recently, data have been presented that a different outcome of NLRP3 activation, loss of *Slc1a3,* alters glutamate and glutamine metabolism leading to reduced clearance of amyloid by microglial phagocytosis [41]. Thus, the relationship between NLRP3 inflammasome activation and enhanced amyloidosis and tauopathy need not passage through IL-1β.

Indeed, we expected a greater induction of IL-1β by tauopathy based on reports from other inflammatory conditions. Recombinant IL-1RA is FDA approved for rheumatoid arthritis, in which IL-1β is increased two-fold in plasma [42] and 100-fold in synovial [43] fluid compared to individuals without rheumatoid arthritis. In mouse models of AD-like pathology, tauopathy alone induces an IL-1β response. Jiang et al. [19] reports an almost two-fold increase in IL-1β from six to nine months of age in rTg4510 mice and Ising et al. [16] report an approximate 50% increase in IL-1β(p17) in Tau22 mice compared to nontransgenic controls. Models of amyloid deposition appear to have an even greater IL-1β response, Heneka et al. [15] report an almost three-fold increase in IL-1β compared to nontransgenic controls. However, in human AD brain

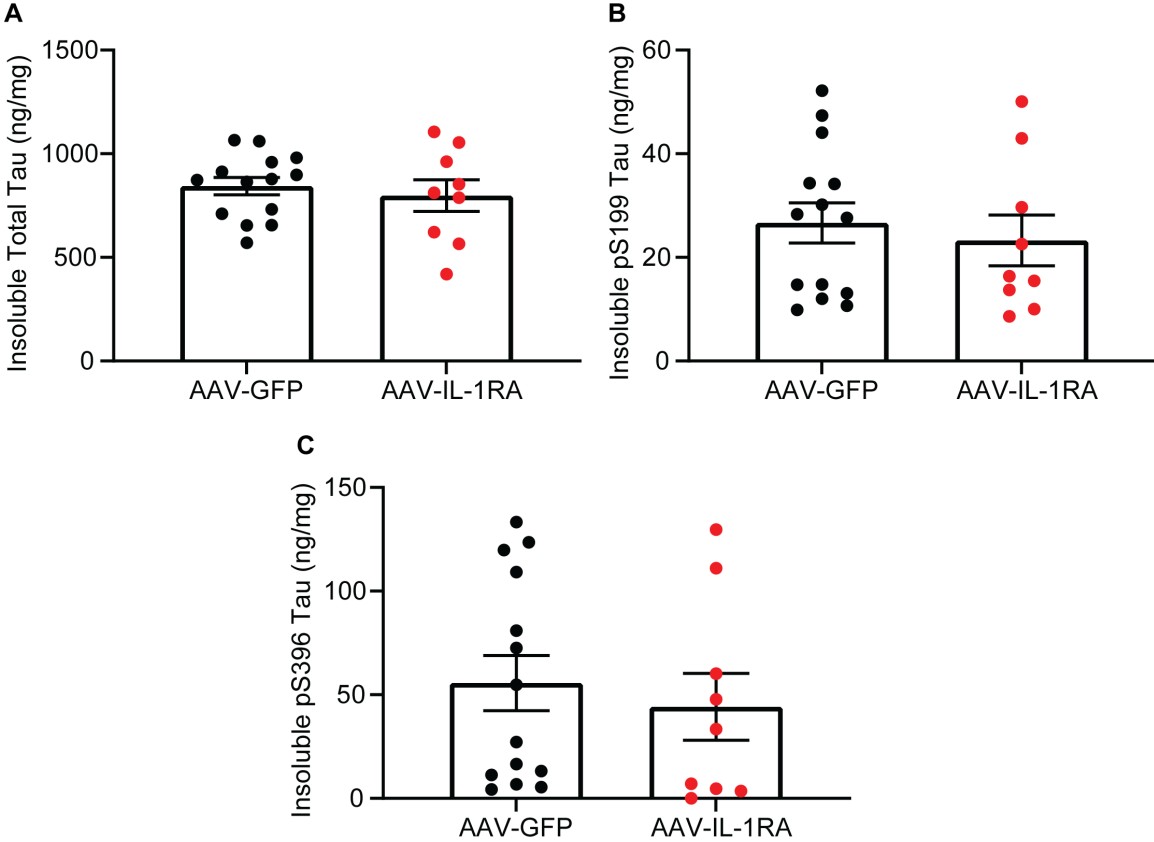

**Fig 8. IL-1RA over-expression does not modify insoluble tau in PS19 mice.** Graph of RIPA-insoluble total tau **(A)**, pS199 tau **(B)**, and pS396 tau **(C)**. IL-1RA over-expression did not modify RIPA-insoluble tau as measured by ELISA. Data normalized to total protein concentration and presented as mean ± SEM; points represent values of individual mice; n = 9-14. Analyzed by two-tailed Student's t-test.

there is an approximately 10-fold increase in IL-1β(p17) compared to control samples greater again than either tauopathy or amyloidosis alone in mouse brain [16]. With respect to the ability of IL-1β to drive tauopathy, Ghosh et al. [10] reported over-expression of IL-1β can increase phospho-tau but these mice had a nine-fold increase in IL-1β over endogenous levels. Therefore, it may be the case that amyloid pathology elicits a greater increase in IL-1β than tauopathy alone and a large increase in IL-1β may be necessary to increase tauopathy.

One interesting observation in our initial experiment in rTg4510 mice pertains to the emergence of insoluble tau deposits. Since there was no effect of IL-1RA over-expression on total or phospho-tau measures by ELISA, we correlated soluble total tau values in the right and left hemispheres and observed a significant correlation. This was expected as total soluble tau should be predominantly influenced by the tau transgene expression which appears to be regionally similar for a given mouse. However, when we correlated insoluble total tau measures in the right and left hemispheres, we did not observe a significant correlation. This implies that tau deposits either initiate, elongate or accumulate in a regionally independent and stochastic manner within mouse brain.

There are limitations to the study and it is important that they be addressed. Our study was conducted at only one time-point of disease in each model. In both studies, the mice were injected after aberrant hyperphosphorylation and deposition of tau had already begun. Therefore, it must be considered that IL-1RA may be more effective earlier in the disease course. Additionally, the route of administration may not have been optimal. A recent study identified IL-1R1 expression in

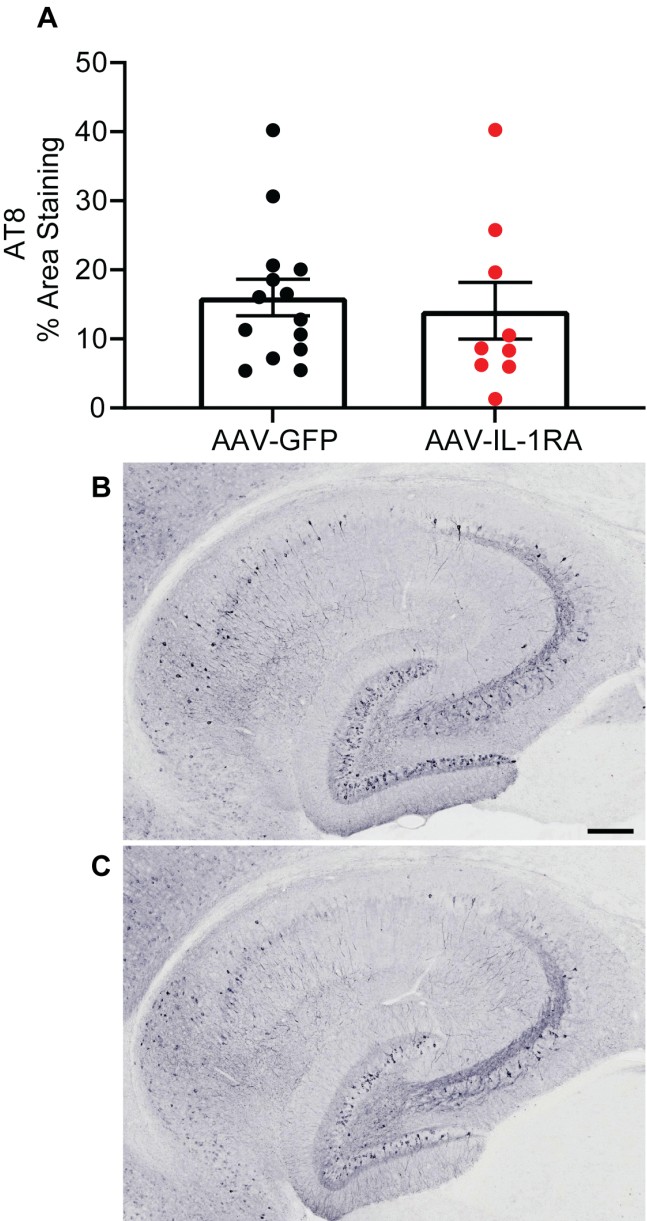

**Fig 9. IL-1RA over-expression does not modify AT8 immunoreactivity in PS19 mice. (A)** Percent area staining in cortex and hippocampus of AT8 tau. IL-1RA over-expression did not reduce AT8 immunoreactivity. **(B & C)** Representative images of AT8 staining in hippocampus of AAV-GFP **(B)** and AAV-IL-1RA **(C)** injected mice. Data presented as mean±SEM; points represent values of individual mice; n=9-14. Analyzed by two-tailed Student's t-test. Scale bar=200 mm.

endothelial cells, ventricular cells, astrocytes, and dentate gyrus neurons but not in microglia. Stimulation of endothelial cells with IL-1 induces expression of IL-1 by microglia [44]. It is possible that it is necessary to interfere in this cascade at the level of the ependymal cells. However, IL-1RA is reported to cross the blood-brain-barrier and IL-1β has been shown to enhance tau phosphorylation even *in vitro*, therefore we would predict that IL-1RA should reduce phospho-tau even in a therapeutic experimental design [13]. Finally, our experiments were designed to test the discrete hypothesis that

**Table 1. Differentially expressed genes identified by bulk RNA sequencing in posterior cortex of IL-1RA injected PS19 mice compared to GFP injected PS19 mice.** Bulk RNAseq analysis reveals differentially expressed genes related to antigen presentation in IL-1RA injected mice compared to GFP injected mice.

| Gene Name | Log$_2$(Fold Change) | FDR-adjusted P-value |
|---|---|---|
| H2-Aa | −2.19 | 0.0026 |
| Cd74 | −2.19 | 0.0049 |
| H2-Ab1 | −1.97 | 0.1015 |
| Nr4a1 | −0.94 | 0.1015 |
| Gm15425 | −0.88 | 0.0003 |
| Slc38a5 | −0.84 | 0.1980 |
| Gm43684 | 1.22 | 0.0003 |

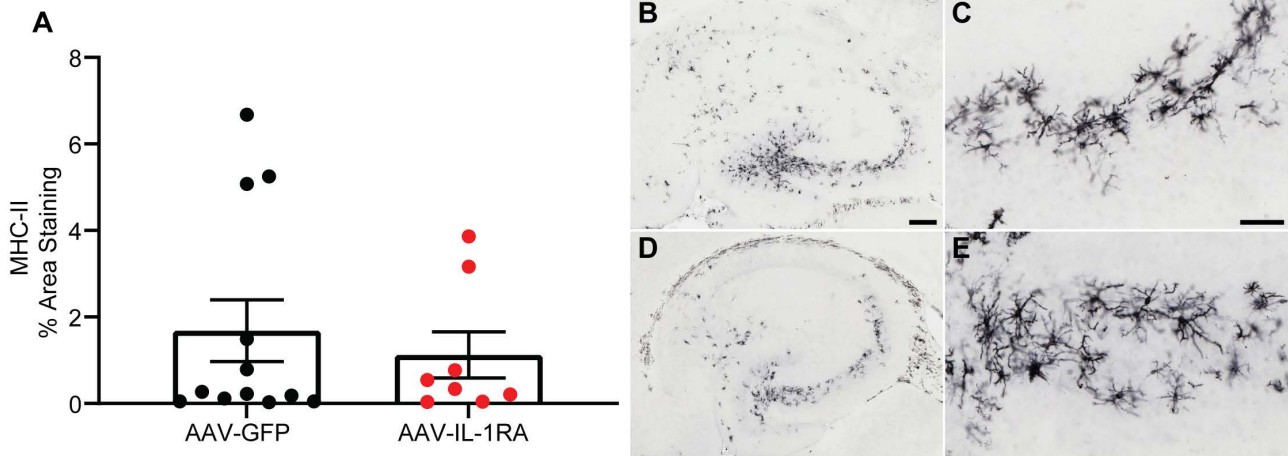

**Fig 10. IL-1RA over-expression does not modify MHC-II immunoreactivity in PS19 mice. (A)** Percent area staining in cortex and hippocampus of MHC-II. IL-1RA over-expression did not reduce MHC-II immunoreactivity. **(B – E)** Representative images of MHC-II staining in hippocampus of AAV-GFP **(B & C)** and AAV-IL-1RA **(D & E)** injected mice. Data presented as mean ± SEM; points represent values of individual mice; n = 8-12. Analyzed by two-tailed Student's t-test. **(B, D)** Scale bar = 200 mm; **(C, E)** Scale bar = 50 μm.

IL-1RA would ameliorate tauopathy based on prior reports in the literature. While we did observe a significant reduction of TOC1-positive tau in rTg4510 mice, holistically, our data do not support this hypothesis. However, we cannot draw broader conclusions about the role of IL-1β or IL-1R1 signaling in Alzheimer's disease. Future experiments using genetic knockouts of either *Il1r1* or *Il1b* in models of AD-like pathology would be instrumental in answering these more mechanistic questions.

In conclusion, our data do not support the use of central supplementation of IL-1RA for the treatment of tauopathies, possibly including Alzheimer's disease. Furthermore, our data suggest that reduced canonical IL-1β signaling may not be responsible for the observed effects of NLRP3 disruption in mouse models of tauopathy. However, additional mechanistic studies must be performed to determine what is mediating the effects of NLRP3 inhibition on tau pathology. The prior study in which an IL-1R blocking antibody, administered systemically, was found to be beneficial was conducted in the 3xTg mouse model, which has both amyloid and tau pathology [11]. Given that the tauopathy in this model appears to be dependent upon amyloidosis [45,46], IL-1β may play a greater role in the stimulation of tauopathy by amyloid than in the development of tauopathy in the absence of amyloid. It is also feasible that the peripheral effects of the anti-IL-1β antibody mediated the effects on brain pathology. While prior studies have demonstrated that IL-1β can exacerbate tauopathy, the

disruption of IL-1 signaling alone does not appear to be sufficient to mitigate tauopathy. Therefore, it may be necessary to interfere with pro-inflammatory signaling more broadly to have a significant impact on tauopathy.

## Supporting information

**S1 Fig. Treatment with IL-1RA does not induce cell death in IL1R1 reporter cell line.** MTT assay shows no significant changes with either IL-1β treatment or IL-1RA treatment.
(TIF)

**S2 Fig. Western blotting of soluble phospho-tau confirms ELISA data.** (A) Graph of pS199/202 phospho-tau detected by Western blot relative to total protein loading control. No significant difference observed between AAV-GFP and AAV-IL-1RA injected mice. Data analyzed by two-tailed Student's t-test. (B) Representative images of pS199/202 phospho-tau (top panel) and total protein loading control (bottom panel). (C) Graph of pS199/202 phospho-tau levels by western correlated with pS199 phospho-tau levels detected by ELISA. A significant correlation was observed between the two methods (p = 0.0008). Red dots represent AAV-IL-1RA injected mice and black dots represent AAV-GFP injected mice. Data presented as mean ± SEM, n = 8–11.
(TIF)

**S3 Fig. Raw uncropped images of western blots in S2 Fig.**
(PDF)

**S1 Dataset. Raw data collected and analyzed in this study.**
(XLSX)

## Acknowledgments

The authors would like to thank the Van Andel Institute Genomics Core for their assistance with RNA sequencing (Grand Rapids, MI; RRID:SCR_022913) and Dr. Robert Vaughan for his guidance in setting up the RNAseq experiment.

## Author contributions

**Conceptualization:** Marcia Gordon.

**Data curation:** Rama Shankar, Bin Chen.

**Formal analysis:** Dylan J. Finneran, Ahlam S. Soliman, Rama Shankar.

**Funding acquisition:** Dave Morgan, Marcia Gordon.

**Investigation:** Dylan J. Finneran, Brianna M. Jackman, Taylor Desjarlais, Alayna Henry, Ahlam S. Soliman, Patricia C. Muskus, Kevin R. Nash, Marcia Gordon.

**Supervision:** Bin Chen, Dave Morgan, Marcia Gordon.

**Visualization:** Dylan J. Finneran.

**Writing – original draft:** Dylan J. Finneran, Rama Shankar, Dave Morgan, Marcia Gordon.

**Writing – review & editing:** Brianna M. Jackman, Taylor Desjarlais, Alayna Henry, Ahlam S. Soliman, Patricia C. Muskus, Bin Chen, Kevin R. Nash.

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
