## [Decision Letter · Decision Letter 0]

29 May 2025

Dear Dr. Gordon,

Thank you for submitting your manuscript to PLOS ONE. After careful consideration, we feel that it has merit but does not fully meet PLOS ONE’s publication criteria as it currently stands. Therefore, we invite you to submit a revised version of the manuscript that addresses the points raised during the review process.

We look forward to receiving your revised manuscript.

Kind regards,

Kai-Hei Tse

Academic Editor

PLOS ONE

Journal Requirements:

4. Please remove all personal information, ensure that the data shared are in accordance with participant consent, and re-upload a fully anonymized data set.

Reviewers' comments:

Reviewer's Responses to Questions

**Comments to the Author**

1. Is the manuscript technically sound, and do the data support the conclusions?

Reviewer #1: Yes

Reviewer #2: No

2. Has the statistical analysis been performed appropriately and rigorously?

Reviewer #1: Yes

Reviewer #2: I Don't Know

3. Have the authors made all data underlying the findings in their manuscript fully available?

Reviewer #1: Yes

Reviewer #2: Yes

4. Is the manuscript presented in an intelligible fashion and written in standard English?

Reviewer #1: Yes

Reviewer #2: Yes

Reviewer #1: The manuscript “Development of mature tau pathology is not dependent on interleukin-1 receptor signaling in two mouse models of tauopathy” by Finneran et al. investigates the role of interleukin-1 receptor (IL-1R) signaling in the progression of tau pathology using two mouse models of tauopathy (rTg4510 and PS19). The authors demonstrate that IL-1 receptor antagonist (IL-1RA) effectively inhibits IL-1β-mediated signaling in vitro. However, overexpression of IL-1RA in vivo failed to decrease tau pathology significantly, leading the authors to conclude that IL-1R signaling does not play a significant role in tau progression in these models (except for reducing TOC1-positive tau oligomers in rTg4510 mice with intracerebral injections of AAV expressing IL-1RA). While the study addresses an essential question regarding neuroinflammation and tauopathy, several aspects of the experimental design and interpretation limit the overall conclusions. The following comments may help improve the manuscript and clarify the findings:

Major comments:

1. Although the authors report approximately 300-fold overexpression of IL-1RA in the ipsilateral cortex of rTg4510 mice and 10-fold overexpression in PS19 mice relative to endogenous levels, they do not assess whether these concentrations are sufficient to inhibit IL-1β in vivo. Prior studies suggest that an IL-1RA:IL-1β ratio of 10–100 is required for effective inhibition (PMID: 2139669, PMID: 1824616, PMID: 2147937). Determining IL-1RA and IL-1β levels and their ratio would help to assess whether functional blocking of IL-1β was achieved in their mouse models.

2. While IL-1RA bioactivity was validated in vitro using a reporter cell line, the authors did not perform in vivo validation of IL-1β signaling inhibition. They could have challenged animals with LPS and assessed downstream inflammatory markers such as IL-6 to confirm effective blocking of IL-1β. Without such confirmation, it is unclear if the overexpression of IL-1RA seen in the mouse models resulted in inhibition of IL-1β in the brain.

3. The manuscript reports elevated Il1b mRNA in both models but does not assess levels of mature IL-1β protein, cleaved caspase-1, NLRP3 inflammasome, or downstream IL-1R signaling components such as NF-κB or MyD88. Without this information, whether IL-1β signaling is active at the ages they tested cannot be determined. It could be that IL-1β plays a more significant role at earlier stages of disease, before the intervention time points done in this study.

4. To understand if IL-1RA over-expression had an impact, IL-1R1 expression levels and localization in the brain, authors are advised to refer to Liu X et al., (PMID: 30893590) in terms of normal physiological concentration (pM) vs pathological concentration (nM) of IL-1b. There could be two possibilities: a) low expression of IL-1R1 in cells other than endothelial and ependymal cells of the brain could explain the lack of response to IL-1RA (per PMID: 30893590) - that means, no point in having 300 or 100 fold over expression of IL-1RA when there is no receptor present in important cells (like microglia during early stages - until primed by endothelial cells?); b) 3 months (in rTg4510) and 6-months (PS19) time points may be already too late to block IL-1R signaling, which seem to originate in endothelial cells first. These points need to be discussed.

5. On a related note, the authors could have considered including two groups and starting IL-1RA both before and after the onset of tau pathology to understand whether IL-1R signaling plays a role in the initiation of disease versus its progression and assess the impact of IL-1RA on both. A short discussion on this limitation would also help.

6. If the authors hypothesize that tau pathology is independent of IL-1R activation, including IL-1β knockdown or knockout experiments in tauopathy models could help confirm this. Or highlighting these in the discussion is essential.

7. Including a brief explanation in the introduction or discussion about the IL-1R ligands (e.g., IL-1α, IL-1β, IL-1RA) and how IL-1RA competes with pro-inflammatory ligands for receptor binding would be helpful.

8. Only one method (ELISA) analyzed total and phosphorylated tau, which is not typical in the field. Some of the findings from ELISA (in Figs 3 and 4) for phosphorylated and total tau are also measured via Quantitative Western Blot.

9. On a related note, it would have been helpful to include a non-transgenic sample to confirm that the pS199/pS396 are indeed higher in rTg4510 mouse brains.

10. Finally, have the authors looked at the IL-1R1 levels in the brains of 3- and 6-month old rTg4510 and PS19 mice, respectively, in relation to non-transgenic controls as a first step before intervening with the IL-1RA?

Minor comments

1. Fig. 1c and 1d – please specify what is measured in the Y axis (instead of ‘optical density’).

2. Fig. 2b – Please provide statistics comparing contralateral groups (AAV-GFP vs AAV-IL-1RA).

Reviewer #2: In this study, the authors overexpressed interleukin-1 receptor antagonist (IL-1RA) in two different

transgenic mouse models (rTG4510 and PS19) of tauopathy. However, the data showed that there

were no significant changes in tauopathy after overexpression of IL- 1RA, so they concluded that

interleukin-1 receptor signaling does not play a significant role inthe progression of tauopathy in

these two transgenic mouse models. Although it seems interesting, the experimental design

lacks rigorous consideration, and the conclusion is not persuasive enough. Attached are some

specific comments:

1. Figure 1A: The authors only check the mRNA level of IL1b; the protein level should also be

detected.

2. Figure 2: The authors should include measuring IL-1β levels after the AAV-1L-1RA

treatment.

3. Figure 2C: The changes are not noticeable here. The authors should use a more

representative image.

4. Figure 3 - 4: Only using ELISA measurement is not sensitive and reliable enough. The authors

should consider using a Western blot to check.

5. Figure5:Theauthor shouldexplainmore aboutwhy solubletau depositioncorrelated with IL1RA overexpression rather than insoluble tau deposition.

6. Figure 6: Still, how about the IL-1β signal levels change after overexpression of IL- 1RA

here? The author should include these measurements.

7. Figure 7-9: Same as Figure 3-4, Western blot is suggested to be used to measure the protein

level changes.

8. Figure 10: The author should explain in detail why MCH-II’s expression is affected by the IL1RA.

**Do you want your identity to be public for this peer review?** For information about this choice, including consent withdrawal, please see our Privacy Policy

Reviewer #1: No

Reviewer #2: No

---

## [Author Response · Author response to Decision Letter 1]

11 Sep 2025

The authors would like to thank the reviewers for the cogent and thoughtful comments and helping to strengthen the manuscript. The authors would like to note that these two experiments were designed to test the hypothesis that using a gene therapy approach to over-express IL-1RA (recombinant IL-1RA is FDA-approved) would reduce tauopathy based on extensive literature in both mouse model systems and human studies implicating IL-1β in the progression of AD. We believe our data are timely and relevant to the current focus on inflammasome activation, which can increase secretion of IL-1β (among other actions). The goal of our experiment was to test this discrete, translational hypothesis, not to fully explore the biology of IL-1β or IL1R1 signaling in tauopathy. Our resulting data do not support the use of IL-1RA to diminish tauopathy. Although this does not directly contradict the possibility that IL-1 plays a role in Alzheimer’s pathogenesis, it would imply use of IL-1RA therapy is unlikely to benefit patients with tauopathy. Therefore, some of the suggested additional experiments we believe are beyond the scope of our conclusions. We have amended the revision to better align our conclusions with our hypothesis and data and to more fully discuss the limitations of our study design.

There were common concerns presented by both reviewers, which we would like to address here first. Both reviewers suggested measures Il1b message was insufficient and asked that we measure IL-1β protein in our model mice to confirm an increase. We have performed an ELISA for IL-1β in the soluble fraction of tissue homogenates of anterior cortices. We observed a significant 75% elevation of IL-1β in 6-month-old rTg4510s compared to 4 month-old nontransgenic mice (Figure 1B). Similarly, we observed small (10%) but significantly elevated IL-1β protein levels in 9-month-old PS19 mice compared to age-matched nontransgenic mice (Figure 6B). Published reports also detect modestly elevated IL-1β in rTg4510 mice by ELISA at 9 months (a trend was observed at 6 months) (Jiang et al., 2021) and PS19 mice at 4 months of age by IHC, preceding the development of mature tauopathy in PS19 mice (Yoshiyama et al., 2007). Using our IL-1β ELISA data, we were able to calculate the ratio of IL-1RA to IL-1β and we achieved approximately 70,000-fold and 1,000-fold excess IL-1RA in rTg4510s and PS19s, respectively. Thus, IL-1β protein is increased in our model systems and over-expression of IL-1RA by AAV yielded levels above the minimally effective IL-1RA to IL-1β ratio in both of our models.

Admittedly, we expected a greater induction of IL-1β by tauopathy based on reports from other inflammatory conditions. Recombinant IL-1RA is FDA approved for rheumatoid arthritis, in which IL-1β is increased by a factor of 2 in plasma (Eastgate et al., 1988) and a factor of 100 in synovial fluid (Kahle et al., 1992) compared to individuals without rheumatoid arthritis. In rTg4510s, Jiang et al. (2021) reports an approximately two-fold increase in IL-1β from 6-9mo, in line with our results. In Tau22 mice there is a reported 50% increase in IL-1β(p17) compared to nontransgenic controls (Ising et al., 2019). In an amyloid mouse model, Heneka et al. (2013) report an almost 3-fold increase in IL-1β compared to nontransgenic mice. However, in human AD brain there is an approximately 10-fold increase in IL-1β(p17) (Ising et al., 2019). Ghosh et al. (2013) reported IL-1 over-expression increased pTau in 3xTG mice, but these mice had a 9-fold increased expression of IL-1 over endogenous levels. It may be the case that amyloid pathology more strongly elicits increases in IL-1β than tauopathy, and this lower IL1β expression does not drive further tauopathy. We have added this information in the discussion

Additionally, both reviewers asked that we confirm our tau ELISA results using western blotting or some other method and to confirm that our tau transgenic mice do indeed have increased phospho-tau compared to nontransgenic mice. We did not run nontransgenic mice on tau ELISAs in our experiment involving rTg4510 mice, because in our 15 years of experience with the rTg4510 transgenic mouse line these mice have massively more phospho-tau and insoluble tau than nontransgenic littermate mice (Dickey et al., 2009; Lee et al., 2010; Brownlow et al., 2013; Nash et al., 2013; Selenica et al., 2013; Brownlow et al., 2014; Selenica et al., 2014a; Selenica et al., 2014b; Delic et al., 2015; Hunt et al., 2015; Joly-Amado et al., 2016; Finneran et al., 2019; Sandusky-Beltran et al., 2019; Joly-Amado et al., 2020). Additionally, the tau ELISA kits we purchased are advertised as using human-tau specific antibodies, (although some cross reactivity with mouse tau can be observed with high concentrations of human-tau antibodies such as AT8). Since we have less experience with the PS19 transgenic mouse line, we ran n=4 nontransgenic mice on each ELISA we performed in the study involving PS19 mice at equivalent dilutions to the transgenic mice and mouse tau was below the limit of detection in our nontransgenic samples. We have elected to use ELISA to measure tau pathology because the standard curves permit absolute quantitation (pg/mg) rather than relative concentration of the analyte (arbitrary units). Nonetheless, we have confirmed the pS199 ELISA data in this study by western with a similar antibody (pS199/202) and observed similar result that correlated well with the ELISA data (see supplemental figures). Additionally, we have also validated these ELISA kits by comparison to Western blots on a separate cohort of animals in a published study (Tetlow et al., 2023). Therefore, we believe our tau ELISA data to be reliable.

Below are our responses (in black) to specific reviewer comments (in red). We thank the reviewers again for their time spent reviewing our manuscript as well as their helpful comments in making this a stronger submission. We hope they find our responses satisfactory and look forward to any additional comments they may have.

Major comments:

2. While IL-1RA bioactivity was validated in vitro using a reporter cell line, the authors did not perform in vivo validation of IL-1β signaling inhibition. They could have challenged animals with LPS and assessed downstream inflammatory markers such as IL-6 to confirm effective blocking of IL-1β. Without such confirmation, it is unclear if the overexpression of IL-1RA seen in the mouse models resulted in inhibition of IL-1β in the brain.

The authors agree that a challenge experiment in which mice are administered AAV-GFP or AAV-IL1RA and then challenged with IL-1β (LPS has many other actions than increasing IL-1) ~4wks later would be a definitive demonstration of in vivo biological activity. However, this would require production of new AAV preparations, development of the mouse IL-1 injection model, and measurements of multiple outcome measures that would exceed the time restriction set for this revision. Additionally, the funding for this project is now expired and we are not able to undertake additional animal experiments. We have indicated this potential limitation of our study in the revised Discussion section but feel reasonably confident in the biological activity of the construct given the in vitro data and levels of in vivo expression of the protein.

3. The manuscript reports elevated Il1b mRNA in both models but does not assess levels of mature IL-1β protein, cleaved caspase-1, NLRP3 inflammasome, or downstream IL-1R signaling components such as NF-κB or MyD88. Without this information, whether IL-1β signaling is active at the ages they tested cannot be determined. It could be that IL-1β plays a more significant role at earlier stages of disease, before the intervention time points done in this study.

The authors thank the reviewer for this important question. Published reports contain increased protein measures of NLRP3, ASC, and cleaved caspase-1 in 6 month old rTg4510 mice (Jiang et al., 2021; de Oliveira et al., 2022). Similar results have been reported in PS19 mice at 3 and 6 months of age (Zhang et al., 2024). Coupled with our new data demonstrating increased IL-1β protein in our mice, we feel confident that IL-1β signaling is occurring in our model animals at the ages at which we applied our treatment. However, as indicated above, this level of increased signaling may not be sufficient to drive additional tau phosphorylation or deposition. This information has been added to the introduction section and our IL-1β ELISA results have been included in a revised figure in the manuscript.

4. To understand if IL-1RA over-expression had an impact, IL-1R1 expression levels and localization in the brain, authors are advised to refer to Liu X et al., (PMID: 30893590) in terms of normal physiological concentration (pM) vs pathological concentration (nM) of IL-1b. There could be two possibilities: a) low expression of IL-1R1 in cells other than endothelial and ependymal cells of the brain could explain the lack of response to IL-1RA (per PMID: 30893590) - that means, no point in having 300 or 100 fold over expression of IL-1RA when there is no receptor present in important cells (like microglia during early stages - until primed by endothelial cells?); b) 3 months (in rTg4510) and 6-months (PS19) time points may be already too late to block IL-1R signaling, which seem to originate in endothelial cells first. These points need to be discussed.

10. Finally, have the authors looked at the IL-1R1 levels in the brains of 3- and 6-month old rTg4510 and PS19 mice, respectively, in relation to non-transgenic controls as a first step before intervening with the IL-1RA?

The authors thank the reviewer for bringing this important study to their attention. We believe our approach is agnostic to which cell-types are expressing IL-1R1. Several studies have demonstrated that IL-1β is elevated in AD and mouse models of AD and that increased IL-1β can exacerbate AD-like pathology in mouse models. Therefore, our hypothesis was that antagonizing IL-1β signaling may be a potential therapeutic strategy in AD and was agnostic as to the precise biological mechanism(s) by which IL-1β may be exacerbating AD-like pathology. We have included the data from Liu et al in our discussion of IL1R1 biology in the Introduction section.

5. On a related note, the authors could have considered including two groups and starting IL-1RA both before and after the onset of tau pathology to understand whether IL-1R signaling plays a role in the initiation of disease versus its progression and assess the impact of IL-1RA on both. A short discussion on this limitation would also help.

6. If the authors hypothesize that tau pathology is independent of IL-1R activation, including IL-1β knockdown or knockout experiments in tauopathy models could help confirm this. Or highlighting these in the discussion is essential.

The authors agree that administering IL-1RA at a single timepoint in the disease, analogous to early in the disease state in humans, is a potential limitation of the study. However, we chose the time points for administering the AAV vectors to be relatively early such that either a role in prevention or in treatment of existing pathology could be detected. Furthermore, we agree that a more complete exploration of the role of IL-1β or IL1R1 signaling in these models would include genetic knockouts of both ligand and receptor. This is important future work that would meaningfully add to the discussion of inflammation in tauopathy and we have included it as a possible future direction in the Discussion section.

7. Including a brief explanation in the introduction or discussion about the IL-1R ligands (e.g., IL-1α, IL-1β, IL-1RA) and how IL-1RA competes with pro-inflammatory ligands for receptor binding would be helpful.

The authors agree that a brief discussion of IL1R1 biology is important context and we have added this to the Introduction section.

Minor comments

1. Fig. 1c and 1d – please specify what is measured in the Y axis (instead of ‘optical density’).

The Y-axis and figure legend has been altered for increased clarity.

2. Fig. 2b – Please provide statistics comparing contralateral groups (AAV-GFP vs AAV-IL-1RA).

All of the statistically significant comparisons are displayed on the graph. Although the two contralateral sides appear to be significantly different, this is likely be due to the use of the log scale Y-axis. The massive increases in IL-1RA in the ipsilateral treated group makes the variance of the overall experiment quite large. However, a direct T-test between the IL1-RA and GFP contralateral sides does demonstrate a significant difference. The figure has been updated to include a “ns” label for clarity.

Reviewer #2: In this study, the authors overexpressed interleukin-1 receptor antagonist (IL-1RA) in two different transgenic mouse models (rTG4510 and PS19) of tauopathy. However, the data showed that there were no significant changes in tauopathy after overexpression of IL- 1RA, so they concluded that interleukin-1 receptor signaling does not play a significant role in the progression of tauopathy in these two transgenic mouse models. Although it seems interesting, the experimental design lacks rigorous consideration, and the conclusion is not persuasive enough. Attached are some specific comments:

Please note the goal of the study was a test of IL-1RA gene therapy (see above), not a direct test of the IL-1 hypothesis. While the results do not support the IL-1 hypothesis, they are not sufficient to eliminate some form of the hypothesis. We have made this more clear throughout the manuscript.

2. Figure 2: The authors should include measuring IL-1β levels after the AAV-1L-1RA

treatment.

6. Figure 6: Still, how about the IL-1β signal levels change after overexpression of IL- 1RA

here? The author should include these measurements.

The authors agree this would be an interesting finding but unfortunately we do not have sufficient sample remaining to perform the measurements. We were able to measure the basal levels in the tau mice using additional samples of the same ages to establish IL-1 was increased, albeit by relatively small amounts (see above).

3. Figure 2C: The changes are not noticeable here. The authors should use a more

representative image.

The authors apologize for the confusion. This figure is IHC for GFP in both panels, which was used as a reporter for AAV-IL1-RA expression. These images are meant to demonstrate similar expression of both AAV constructs in mouse brain.

5. Figure5: The author should explain more about why soluble tau deposition correlated with IL1RA overexpression rather than insoluble tau deposition.

The authors apologize for the confusion. What is being correlated in these two figures are soluble total tau in the left hemisphere and soluble total tau in the right hemisphere, which are very tightly correlated, suggesting common causes for the interindividual animal variances (such as transgene copy number). However, insoluble total tau in the left hemisphere is not significantly correlated with insoluble total tau in the right hemisphere, indicating that deposition of tau may occur in a stochastic manner even within the same brain of an animal, with one side initiating deposition at a different time than the other.

8. Figure 10: The author should explain in detail why MCH-II’s expression is affected by the IL1RA.

We probed for MHC-II based on the results of our RNAseq experiment. Several of the differentially regulated genes are involved in antigen presentation and we wanted to confirm a reduction at the protein level. Reductions in these genes are broadly consistent with our predicted anti-inflammatory effect of IL-1RA and a reduction in MHC-II associated proteins has been observed after anti-inflammatory treatment by others in this mouse model (Yoshiyama et al.). We have made the rationale for this more clear in our Discussion section.

In Summary, we have been able to respond to some of the reviewer’s requests with new data, in some cases using archival samples from PS19 and rTg4510 mice. However, the additional animal experiments are beyond the scope of this manuscript as they would require 1) Generation of additional lots

---

## [Editor Report · Decision Letter 1]

18 Sep 2025

Dear Dr. Gordon,

Thank you for submitting your manuscript to PLOS ONE. After careful consideration, we feel that it has merit but does not fully meet PLOS ONE’s publication criteria as it currently stands. Therefore, we invite you to submit a revised version of the manuscript that addresses the points raised during the review process.

We look forward to receiving your revised manuscript.

Kind regards,

Suhail Rasool

Academic Editor

PLOS ONE

Journal Requirements:

Additional Editor Comments:

The authors have concluded interleukin-1 receptor signaling does not play a significant role in the progression of tauopathy using two transgenic models rTg4510 and PS19 models. Both these models does not inherently manifest amyloid-beta (Aβ) pathology; it primarily develops tau pathology, such as neurofibrillary tangles, mimicking human tauopathies. Previously it has been investigated by various researcheres  the role of IL-1β in AD pathogenesis, using  inducible model of sustained IL-1β overexpression using 3xTg-AD model,  tau phosphorylation despite an ∼70–80% reduction in amyloid load and fourfold to sixfold increase in plaque-associated microglia, as well as evidence of greater microglial activation at the site of inflammation. There was also evidence of increased p38 mitogen-activated protein kinase and glycogen synthase kinase-3β activity, which are believed to contribute to tau phosphorylation.

Can the authors comment on that what relvance they can emphsize on their studies. Also does the authors looked over to to pathways like regulation of  p38 mitogen-activated protein kinase and glycogen synthase kinase-3β activity .

---

## [Author Response · Author response to Decision Letter 2]

9 Oct 2025

Revision of Mature tau pathology is not improved by interfering with interleukin-1 receptor signaling in two mouse models of tauopathy

Dear Dr Rasool,

In your response to our revision, you mention the publication by Ghosh et al (2013) examining the role of IL-1 overexpression on tau pathology in the 3xTg mouse model (a paper we have referenced here and on multiple other occasions). The data presented in Ghosh and a few other publications led to our hypothesis that treatment with an FDA approved drug, Anakinra (human IL-1RA) might reduce tauopathy including that in AD. Prior work found elevated IL-1β RNA in the pure tauopathy models used here, and we assumed they would be useful models to test the role of IL-1RA gene therapy as a possible therapy for tauopathies. In spite of achieving high levels of IL-1RA, we failed to detect any impact on tau pathology in these mice. We then went to measure the levels of IL-1β protein in these mice and found relatively small elevations. Thus, we conclude that IL-1β is unlikely to be driving tau pathology in these mice and this explains why the IL-1RA gene therapy had no effects on tauopathy.

Perhaps had we had a mouse with both amyloid and tau, we would have found a different outcome, but our recent experience with the 3xTg model found no tauopathy in aged mice as the mouse has lost phenotype over time (this is also stated to be the case on the JAX web site).

For the most part this was indicated in the paper discussion, but the abstract was still focusing on the role in AD, which we now agree was not fully addressed in the pure tauopathy models. We have rewritten the abstract to more clearly explain the conclusions we have reached. However, for the pure tauopathies, we believe the manuscript explains that IL-1β is unlikely to be driving tauopathy in these models, and thus IL-1RA gene therapy is unlikely to have therapeutic value.

With respect to your second comment, regarding possible alterations in tau kinase regulation due to IL-1RA over-expression, we did not investigate this possibility because we did not observe changes in phospho-tau levels at sites known to be phosphorylated by these kinases. While it is possible that there are changes in the activity of one of these kinases and compensatory increases in activity of the other, our initial hypothesis was that IL-1RA over-expression would reduce tau phosphorylation. Finding that the data did not support this hypothesis, we did not pursue mechanism further.

We now include the revised manuscript, both with and without mark up, indicating the changes in the abstract and in a few places of the discussion.

---

## [Editor Report · Decision Letter 2]

13 Oct 2025

Mature tau pathology is not improved by interfering with interleukin-1 receptor signaling in two mouse models of tauopathy

PONE-D-25-11909R2

Dear Dr. Gordon N. 

We’re pleased to inform you that your manuscript has been judged scientifically suitable for publication and will be formally accepted for publication once it meets all outstanding technical requirements.

Kind regards,

Suhail Rasool

Academic Editor

PLOS ONE
---

## [Editor Report · Acceptance letter]

PONE-D-25-11909R2

PLOS ONE

Dear Dr. Gordon,

I'm pleased to inform you that your manuscript has been deemed suitable for publication in PLOS ONE. Congratulations! Your manuscript is now being handed over to our production team.

Kind regards,

on behalf of

Dr. Suhail Rasool

Academic Editor

PLOS ONE